# SONAR: Spectral-Contrastive Audio Residuals for Generalizable Deepfake Detection

**Ido Nitzan Hidekel** [1]  **Gal Lifshitz** [1]  **Khen Cohen** [2]  **Dan Raviv** [1]

## Abstract

Deepfake audio detectors often fail to generalize to unseen attacks, in part due to *spectral bias*: neural networks prioritize low-frequency structure while under-exploiting subtle high-frequency (HF) artifacts left by generative models. We introduce **SONAR** (Spectral-cONtrastive Audio Residuals), a frequency-guided framework that *explicitly enforces representation-level consistency* between semantic content and HF residuals. Unlike prior frequency-aware or dual-stream detectors that treat HF cues as auxiliary features, SONAR encourages structured interaction between content and noise representations in latent space. The model employs a dual-path architecture in which an XLSR encoder captures low-frequency content, while a parallel branch with learnable, value-constrained 1D SRM (Spatial Rich Model) high-pass filters distills HF residuals. The two representations are fused via frequency cross-attention and trained with a *Jensen–Shannon alignment loss* that promotes LF–HF consistency for genuine audio and amplifies inconsistency for deepfakes. Evaluated on ASVspoof 2021 and in-the-wild benchmarks, SONAR achieves state-of-the-art performance in a **single run** setting and converges faster than strong baselines. By mitigating the effects of spectral bias through frequency-guided alignment, SONAR provides a fully data-driven and architecture-agnostic approach to generalizable audio deepfake detection.

---
[1]School of Electrical Engineering, Tel Aviv University, Tel Aviv, Israel [2]School of Physics and Astronomy, Tel Aviv University, Tel Aviv, Israel. Correspondence to: Ido Nitzan Hidekel <idon@mail.tau.ac.il>, Gal Lifshitz <lifshitz@mail.tau.ac.il>, Khen Cohen <khencohen@mail.tau.ac.il>, Dan Raviv <darav@tauex.tau.ac.il>.

*Proceedings of the 43rd International Conference on Machine Learning*, Seoul, South Korea. PMLR 306, 2026. Copyright 2026 by the author(s).

## 1. Introduction

Generative AI now enables the creation of photorealistic images, video, and speech. In 2024, political deepfakes flooded social media during global elections, while voice-cloning scams caused multimillion-dollar losses, including a 25M$ transfer (United Nations Development Programme, 2024; TRM Labs, 2025). The FBI has warned of AI-powered phishing attacks (Cybersecurity Dive, 2025). More broadly, synthetic media erodes trust in journalism, markets, and legal evidence, making robust and generalizable detection essential.

Most forensic research still centers on increasingly deep classifiers, while largely overlooking how deepfake artifacts disrupt the *joint* statistics of semantic content and noise. Early SRM-style detectors rely on fixed high-pass filters (Fridrich & Kodovský, 2012; Qian et al., 2020) or, in the case of Bayar and Stamm's constrained convolution (Bayar & Stamm, 2016), learnable prediction-error kernels that deliberately suppress content. However, these methods operate on high-frequency (HF) residuals *in isolation*, ignoring the cross-frequency consistency of the underlying signal.

Subsequent two-stream approaches partially address this limitation by introducing a parallel content branch. Han et al. (Han et al., 2021) combine learnable SRM filters with a content pathway, but the two streams are fused only at the output, and no constraint enforces statistical coupling during training. Zhu et al. (Zhu et al., 2024) similarly amplify noise cues for image forgery detection, yet treat them as auxiliary signals and require pixel-level supervision. As a result, none of these approaches model the higher-order consistency between semantic content and HF noise - an interplay that isolated filtering or late fusion cannot capture.

Across both image and audio domains, these approaches share a common structural limitation: high-frequency cues are either suppressed, isolated, or aggregated, but never coupled to semantic content during training. SRM-based methods treat HF residuals as prediction errors to be filtered (Bayar & Stamm, 2016; Fridrich & Kodovský, 2012; Qian et al., 2020), while two-stream architectures (Han et al., 2021; Zhu et al., 2024) introduce parallel content and noise pathways without enforcing statistical relation beyond late fusion. As a result, the natural low–high frequency co-

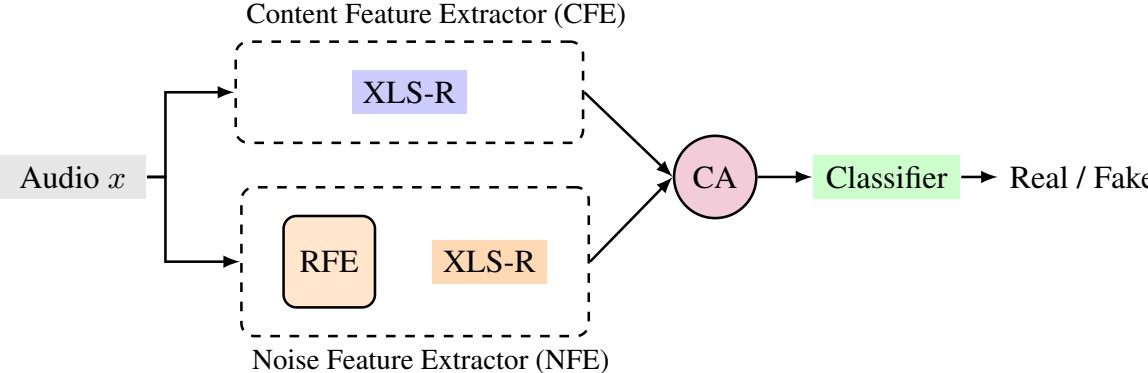

*Figure 1.* **SONAR overview.** Audio is processed in parallel by the Content Feature Extractor (CFE) and the Noise Feature Extractor (NFE). Their embeddings are fused via cross-attention (CA) and classified as real/fake.

modulation present in real signals is not explicitly modeled, and its breakdown in deepfakes remains unexploited.

In audio forensics, the field has progressed from handcrafted spectral features with GMM or LCNN backends (Yamagishi et al., 2019) to fine-tuned self-supervised encoders such as HuBERT and XLSR (Hsu et al., 2021; Tak et al., 2022b; Zhang et al., 2024; Xiao & Das, 2024). While these models achieve strong performance, they remain frequency-agnostic and do not explicitly exploit the structure of HF artifacts. Crucially, neither image-based nor audio-based detectors model the *alignment* between semantic content and high-frequency residuals.

We argue that these shortcomings stem from a common cause: **spectral bias**, also known as the *frequency principle* (F-principle) (Rahaman et al., 2019; Basri et al., 2019; Cao et al., 2021; Xu et al., 2024; Fridovich-Keil et al., 2022), whereby neural networks preferentially learn low-frequency structure while leaving subtle HF cues under-represented. Although some image-based methods route HF residuals through a separate branch, they stop at *separation*. They never model how low- and high-frequency information *should co-vary* in the same feature space during learning.

As a result, no existing detector visual or auditory - actively *aligns* genuine content–noise pairs while *repelling* their fake counterparts in latent space. This *alignment gap* is particularly detrimental in audio, where HF artifacts are easily obscured by perceptual post-processing and bandwidth limitations.

We conduct a statistical analysis to validate the presence of spectral bias in generated speech. We observe that spoofed audio differs from bona-fide speech in higher-order frequency statistics, including a collapse of low–high frequency co-modulation and a systematic shift in low/high energy contrast (Fig. 2a, Fig. 2b). This motivates modeling and aligning low- and high-frequency components at the distributional level.

**Our approach: SONAR and the gap it fills.** We address this gap with **SONAR**, a frequency-guided dual-path framework that *explicitly enforces alignment* between semantic content and high-frequency residuals. SONAR learns a bank of data-driven SRM filters to extract HF residuals and employs a Jensen–Shannon divergence objective to pull content–noise pairs together for genuine audio while pushing them apart for deepfakes. By elevating HF residuals from a nuisance into a supervisory signal and aligning them with semantic content in latent space, SONAR directly counteracts spectral bias, accelerates convergence, and achieves state-of-the-art performance on both controlled benchmarks (ASVspoof 2021) and challenging in-the-wild audio.

To our knowledge, SONAR is the *first* audio deepfake detector to exploit **learnable, distributional alignment** between low- and high-frequency representations.

**Our contributions are threefold:**

- **Frequency-guided alignment of content and noise.** We introduce SONAR, an audio deepfake detector that enforces consistency between low-frequency semantic content and high-frequency residuals in latent space, mitigating spectral bias through data-driven alignment.

- **Spectral bias and generalization failure.** We provide empirical evidence that spectral bias contributes to poor generalization in existing audio deepfake detectors, which overfit low-frequency structure while under-utilizing high-frequency artifacts.

- **State-of-the-art performance with efficient training.** SONAR achieves state-of-the-art Equal Error Rates on ASVspoof2021 (Liu et al., 2023) and In-The-Wild (Müller et al., 2022) benchmarks in a single-run setting, converging in as few as 12 epochs and remaining robust to codec and bandwidth shifts.

**Conflict of Interest Disclosure.** The authors declare no financial conflicts of interest related to this work. The pre-trained models, datasets, and codecs evaluated in this paper are publicly available research artifacts; we have no commercial relationship with any of their developers, and no funding for this work was provided by an entity that builds or sells deepfake detection systems.

## 2. Related Work

**High frequency cues in deep learning.** Fourier features markedly reduce spectral bias in MLPs (Tancik et al., 2020). Successors such as Wave NN (Yang et al., 2022), BiHPF (Jeong et al., 2022), and ADD (Woo, 2022) insert explicit high pass branches or filters, showing that frequency-aware modules consistently sharpen detail capture.

**Frequency domain image forgery detection.** Two stream, high pass pipelines detect manipulation artifacts by pairing low pass content with residual branches (Masi et al., 2020; Qian et al., 2020; Bayar & Stamm, 2016; Fridrich & Kodovský, 2012). Denoising-guided schemes (Zhu et al., 2024) and compact frequency blocks (Tan et al., 2024) further improve generalization with fewer parameters.

**Audio deepfake detection.** Classic systems combine hand-crafted cepstral features with GMM/LCNN backends (Yamagishi et al., 2019). Modern approaches leverage SSL encoders (HuBERT, Wav2Vec, XLSR, Whisper, WavLM) (Hsu et al., 2021; Baevski et al., 2020; Babu et al., 2022; Radford et al., 2023; Chen et al., 2022), yet, as shown by Müller et al. (Müller et al., 2022), they often fail to generalize to unseen architectures. Tak *et al.* fine tuned XLSR with an AASIST head and augmentation for strong OOD results (Tak et al., 2022b), later work fused XLSR layers with specialized classifiers to push performance further (Zhang et al., 2024; Truong et al., 2024; Xiao & Das, 2024). We identify spectral bias as the root cause: these models overfit to robust low-frequency patterns while ignoring the high-frequency artifacts that distinguish neural synthesis.

## 3. Mathematical and Empirical Motivation

Let $x \in L^2(\mathbb{R})$ denote a speech signal and $y \in \{0, 1\}$ its label ($y = 1$: bona fide, $y = 0$: spoofed). We decompose $x$ into complementary frequency components

$$x_L := \mathcal{P}_L x, \qquad x_H := \mathcal{P}_H x, \qquad x = x_L + x_H,$$

where $\mathcal{P}_L$ and $\mathcal{P}_H$ are linear time-invariant projections with disjoint low- and high-frequency support.

**Empirical motivation.** Our statistical analysis (Fig. 2) shows that spoofed speech differs from bona fide speech not only in marginal high-frequency (HF) energy, but in joint structure: (i) a collapse of low–high frequency (LF–HF)

co-modulation and (ii) a systematic shift in LF/HF energy contrast. These effects persist across datasets and codecs, indicating that spoofing disrupts the natural *consistency* between LF structure and HF residuals, rather than merely altering HF magnitude.

**Model hypothesis: LF–HF consistency shift.** We therefore posit that class information is encoded in how high-frequency components co-vary with low-frequency content:

$$\boxed{\mathcal{P}(x_H \mid x_L, y = 1) \neq \mathcal{P}(x_H \mid x_L, y = 0).} \quad (1)$$

Rather than claiming to estimate these conditional distributions, we interpret (1) as a modeling principle: genuine speech exhibits structured LF–HF consistency, while spoofed speech systematically disrupts this relationship. This motivates learning representations that expose LF–HF interactions instead of treating HF information marginally.

**High frequencies as derivatives and residual operators.** For $k \geq 1$, $\widehat{D^k x}(\omega) = (i\omega)^k \hat{x}(\omega)$, so derivatives amplify large $|\omega|$ and suppress low frequencies. Discrete finite-difference and prediction-error operators therefore act as high-pass filters with vanishing response at $\omega = 0$. SRM filters (Fridrich & Kodovský, 2012) realize this principle via short zero-sum kernels ($\sum_t w[t] = 0$), enforcing derivative-like sensitivity to high-frequency residuals.

**Dual representations.** Under (1), neither LF nor HF information alone is sufficient for discrimination. We therefore consider two representations,

$$Z_{\text{content}} = f_L(x_L), \qquad Z_{\text{noise}} = f_H(r(x)), \quad (2)$$

where $r(x)$ denotes SRM-based residual extraction. Separate mappings allow $Z_{\text{noise}}$ to retain derivative-level sensitivity that would otherwise be attenuated by low-frequency–dominated gradients, while $Z_{\text{content}}$ captures semantic structure.

**Distributional consistency via Jensen–Shannon divergence.** To operationalize LF–HF consistency in representation space, we treat $Z_{\text{content}}$ and $Z_{\text{noise}}$ as random variables and compare their empirical distributions using the Jensen–Shannon (JS) divergence, $\text{JS}(Z_{\text{content}}, Z_{\text{noise}})$. JS divergence is symmetric and bounded, providing a stable surrogate for representation-level alignment rather than an estimator of conditional dependence.

We impose the label-conditioned objective

$$\begin{aligned} \mathcal{L}_{\text{align}}(x, y) = {} & y \, \text{JS}(Z_{\text{content}}, Z_{\text{noise}}) \\ & + (1 - y)[1 - \text{JS}(Z_{\text{content}}, Z_{\text{noise}})], \end{aligned} \quad (3)$$

which enforces LF–HF consistency for bona fide speech and amplifies LF–HF inconsistency for spoofed speech. Unlike

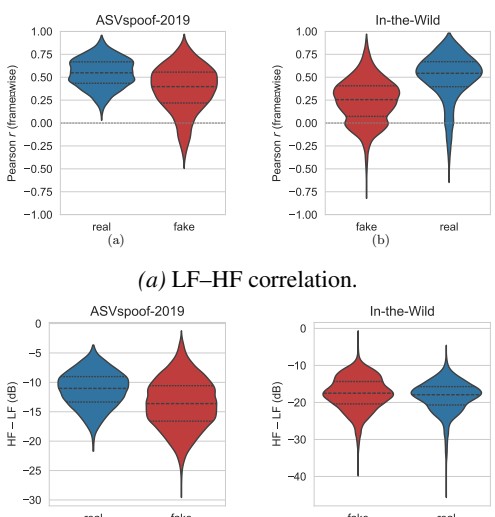

*(a)* LF–HF correlation.

*(b)* HF/LF energy contrast.

*Figure 2.* **Low–high frequency structure reveals spoofing arti-facts.** (a) Pearson correlation between low- (0–4 kHz) and high-frequency (7–8 kHz) bands shows real speech with strong co-modulation ($r \approx 0.6$), while fakes collapse toward zero or negative values. (b) The energy difference $\Delta E = E_{\text{HF}} - E_{\text{LF}}$ is system-atically shifted for fakes across corpora, exposing a consistent HF/LF imbalance. These second-order cues motivate SONAR's *distributional alignment* objective.

pointwise similarity measures (e.g., cosine or $\ell_2$ distance), this objective operates at the level of distributional structure, directly reflecting the LF–HF consistency hypothesis in (1).

## 4. Methodology

**From mathematical motivation to implementation.** Section 3 models audio deepfake detection as a problem of altered low–high frequency consistency, formalized by the conditional shift in (1). Operationalizing this model requires: (i) an explicit construction of high-frequency resid-uals that approximate derivative operators, (ii) separate rep-resentations for low-frequency content and high-frequency residual structure, and (iii) a training objective that acts on their *distributional consistency* rather than pointwise simi-larity.

SONAR instantiates these requirements as follows. First, we extract residual signals $r(x)$ using a constrained, learnable SRM filter bank, implementing the derivative-like operators motivated in Sec. 3. Second, we encode content and resid-ual signals with separate encoders to obtain embeddings $\mathbf{z}_{\text{content}}$ and $\mathbf{z}_{\text{noise}}$, preserving their complementary sensitiv-ities. Finally, we enforce the label-conditioned alignment objective (3), which minimizes divergence for bona fide speech and maximizes it for spoofed speech, thereby restor-ing or amplifying LF–HF consistency in accordance with

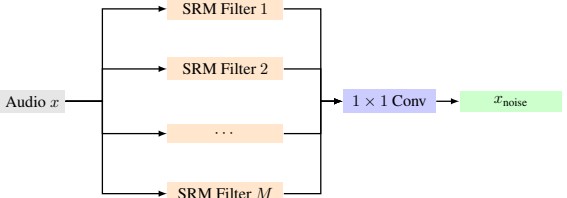

*Figure 3.* **Rich Feature Extractor (RFE).** Audio $x$ is processed by a bank of $M$ SRM-inspired filters, concatenated, and passed through a $1 \times 1$ learnable convolution layer to produce the noise residual representation $x_{\text{noise}}$.

the model.

### 4.1. SONAR Feature Extraction

The SONAR architecture consists of:

- **Content Feature Extractor (CFE):** A Wav2Vec2.0 XLSR (Babu et al., 2022) Encoder.

- **Noise Feature Extractor (NFE):** A module based on constrained SRM filters followed by a Wav2Vec2.0 XLSR Encoder.

- **Fusion via Cross-Attention:** This merges the two feature streams into a unified representation.

Figure 1 illustrates the overall system.

#### 4.1.1. CONTENT FEATURE EXTRACTION (CFE)

Given an input signal $\mathbf{x} \in \mathbb{R}^T$, we extract content features as:

$$\mathbf{z}_{\text{content}} = \text{CFE}(\mathbf{x}) \in \mathbb{R}^{F \times D}, \qquad (4)$$

with $F$ time steps and feature dimension $D$ (encoder depen-dent).

#### 4.1.2. NOISE FEATURE EXTRACTION (NFE)

The Noise Feature Extractor (NFE) builds on a Rich Feature Extraction (**RFE**) module (Figure 3). This module lever-ages constrained SRM filters to emphasize high-frequency components, which are then processed by the Wav2Vec2.0 XLSR Encoder (Babu et al., 2022). The XLSR encoder weights were *not shared* across branches, allowing the model to learn disentangled representations of noise and content. A discussion of the resulting computational cost is provided in section 5.3 and in Appendix G.

**Constrained SRM Filters:** We initialize $M$ (hyperparam-eter for number of filters) learnable filters, each of length 5,

with two key constraints:

$$w_i[m] = -1, \quad \text{(central coefficient)} \quad (5)$$

$$\sum_k w_i[k] = 0, \quad \text{(zero-sum constraint)} \quad (6)$$

where $w_i[m]$ is the $i$-th filter at index $m$ and we initialize the weights from $N(0, I)$. To ensure these are *hard constraints*, after every optimizer step we project the filters back to the constraint set: each filter is divided by the negative of its center coefficient (fixing the middle entry to $-1$), and then its mean is subtracted to enforce strict zero-sum. This guarantees that the constraints hold exactly throughout training without requiring reparameterization or relaxation. These enforced constraints ensure that each filter acts as a high-pass operator, suppressing low-frequency (content) structure while emphasizing high-frequency residuals, consistent with prior work (Bayar & Stamm, 2016; Zhu et al., 2024; Han et al., 2021). Given an input signal $\mathbf{x}$, convolution with the constrained SRM filter bank yields

$$\mathbf{F}_{\text{noise}} = \text{Conv1D}_{\text{SRM}}(\mathbf{x}), \quad (7)$$

where $\mathbf{F}_{\text{noise}} \in \mathbb{R}^{M \times T}$ denotes intermediate residual feature maps. A subsequent learnable $1 \times 1$ convolution without nonlinearity projects $\mathbf{F}_{\text{noise}}$ back to $\mathbb{R}^{1 \times T}$, implementing a linear transformation over the residual channels. This linear mixing preserves high-frequency structure while allowing data-driven reweighting of residual responses for downstream modeling.

### 4.1.3. FUSION AND CLASSIFICATION

The content and noise embeddings, $\mathbf{z}_{\text{content}}$ and $\mathbf{z}_{\text{noise}}$, are fused using a cross-attention mechanism. We conducted all experiments with an embedding dimension of 1024 and 8 attention heads.

$$\mathbf{e}_{\text{out}} = \text{CA}(\mathbf{z}_{\text{content}}, \mathbf{z}_{\text{noise}}) \in \mathbb{R}^{F \times D}. \quad (8)$$

The fused representation is then fed to the AASIST classifier (Jung et al., 2022):

$$\hat{y} = \text{AASIST}(\mathbf{e}_{\text{out}}) \in \mathbb{R}^2, \quad (9)$$

where $\hat{y}$ is the score vector for the real/fake decision.

The architecture of XLSR and AASIST models are detailed in the appendix.

### 4.2. Training Objective

#### 4.2.1. JS-BASED LOSS FOR REAL VS. SPOOF EMBEDDED FREQUENCY DISTRIBUTIONS

Let $\mathbf{Z}_{\text{content}} \in \mathbb{R}^{F \times D}$ and $\mathbf{Z}_{\text{noise}} \in \mathbb{R}^{F \times D}$ denote the content and noise embeddings extracted from the dual path feature

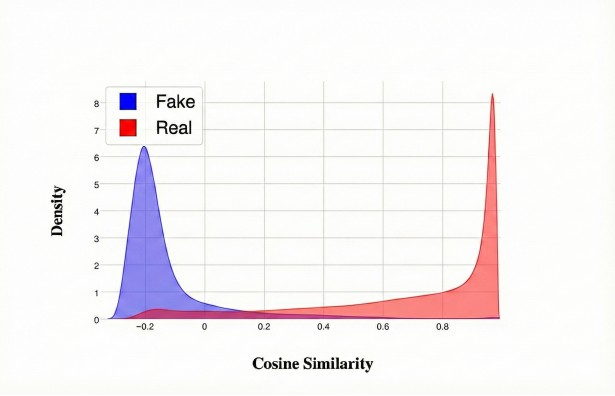

*Figure 4.* **Cosine similarity between LF and HF embeddings.** Real samples cluster near 1, indicating strong LF–HF alignment, whereas fake samples are centered around $-0.2$, reflecting significant dissimilarity between frequency representations.

extractor for an audio example $\mathbf{x}$. Our goal is to increase the probability distance between the frequency embeddings of fake data while simultaneously reducing the distance among those of real data.

To achieve this, we employ the Jensen–Shannon (JS) divergence (Fuglede & Topsøe, 2004) as a metric for comparing distributions. First, since the encoder (XLSR) produce temporal features (2D arrays), we apply a frame-wise softmax to each embedding, per timestamp, converting them into probability distributions for each timestamp. Then, we compute a single JS divergence score, $\text{JS}(\mathbf{z}_{\text{content}}, \mathbf{z}_{\text{noise}})$, using $\log_2$ as the logarithmic base. This ensures that the divergence is normalized to the range $[0, 1]$, facilitating stable comparison across samples.

**Frame-wise JS Divergence:** At each frame $i$, we can treat $\mathbf{z}_{\text{content}}[i]$ and $\mathbf{z}_{\text{noise}}[i]$ as two embeddings in $\mathbb{R}^D$. By applying softmax to each and get two discrete distributions $\mathbf{p}_{\text{content}}[i]$ and $\mathbf{p}_{\text{noise}}[i]$, where the final score for audio $\mathbf{x}$ with embeddings $\mathbf{Z}_{\text{content}}, \mathbf{Z}_{\text{noise}}$ will be:

$$\text{JS}(\mathbf{Z}_{\text{content}}, \mathbf{Z}_{\text{noise}}) = \frac{1}{F} \sum_{i=1}^{F} \text{JS}(\mathbf{p}_{\text{content}}[i], \mathbf{p}_{\text{noise}}[i])$$

We label the example with $y = 1$ if it is *real*, and $y = 0$ if it is *spoof*.

**JS-Based Loss:**

With these settings, we define the custom loss function for each sample $(x, y)$:

Where $\mathbf{Z}_{\text{content}}, \mathbf{Z}_{\text{noise}}$ are the embeddings from our **SONAR** feature extraction.

$$L_{\text{JS}}(x, y) = y \cdot \text{JS}(\mathbf{z}_c, \mathbf{z}_n) + (1 - y) \cdot (1 - \text{JS}(\mathbf{z}_c, \mathbf{z}_n)) \quad (10)$$

We combine the above $L_{\text{JS}}$ (align loss as in equation 3)

*Table 1.* EER (%) on ASVspoof 2021 LA, DF, and In The Wild datasets. Bold entries are best per column. Results from prior work are single-run values, SONAR variants report best (mean) over 3 runs. Statistical significance: SONAR-Full vs. AASIST on ITW ($t = 19.4$, $p = 0.0026$) and SONAR-Finetune vs. XLSR-Mamba on ITW ($t = 4.73$, $p = 0.0419$) both confirm robust improvements ($p < 0.05$).

| Model | LA↓ | DF↓ | ITW↓ |
|---|---|---|---|
| WavLM-Large+MFA (Guo et al., 2024) | 5.08 | 2.56 | – |
| XLSR+AASIST (Tak et al., 2022b) | 1.90 | 3.69 | 10.46 |
| XLSR+MoE (Wang et al., 2024) | – | – | 9.17 |
| XLSR+Conformer (Rosello et al., 2023) | 0.97 | 2.58 | 8.42 |
| XLSR+Conformer+TCM (Truong et al., 2024) | 1.18 | 2.25 | 7.79 |
| XLSR-SLS (Zhang et al., 2024) | 2.87 | 1.92 | 7.46 |
| XLSR-Mamba (Xiao & Das, 2024) | **0.93** | 1.88 | 6.71 |
| **SONAR**-Lite (M=30, $\lambda_{JS} = 1$) | 1.78 (2.03) | 2.11 (2.5) | 6.98 (7.2) |
| **SONAR**-Full (M=30, $\lambda_{JS} = 1$, enhancing (Tak et al., 2022b)) | 1.55 (1.68) | 1.57 (1.95) | 6.00 (6.8) |
| **SONAR**-Finetune (M=30, $\lambda_{JS} = 1$) on (Xiao & Das, 2024) | 1.20 (1.30) | **1.45** (1.62) | **5.43** (5.8) |

with a weighted cross-entropy (WCE) loss, which handles the real/fake classification in a more conventional way and accounts for class imbalance:

$$\mathcal{L}(x, y) = \text{WCE}(\hat{y}, y) + \lambda_{JS} \cdot L_{JS}(\mathbf{z}_{\text{content}}, \mathbf{z}_{\text{noise}}) \quad (11)$$

where $\hat{y} \in [0, 1]$ is the classifier's predicted probability of being real. The scalar $\lambda_{JS}$ balances how strongly the network must enforce the JS-based criterion. After ablation study 5.3, we chose to be $\lambda_{JS}$=1. We emphasize that this loss does not estimate mutual information or conditional dependence, but acts as a stable, bounded regularizer promoting LF–HF consistency.

### 4.3. Alternative Configurations: Lite & Finetune

To validate the intrinsic quality of our representations and demonstrate integration flexibility, we propose two variants. **SONAR-Lite** replaces the complex AASIST head with a simple two-layer MLP fed by mean-pooled dual-path embeddings. Despite this simplification, it maintains competitive performance (Table 1), proving that SONAR's frequency-guided features are robustly discriminative on their own without sophisticated classifiers. **SONAR-Finetune** addresses training costs by injecting our NFE module into the pre-trained XLSR-Mamba pipeline (Xiao & Das, 2024). By treating the original frozen XLSR as the content branch and updating only the noise path and fusion layers, we incorporate frequency guidance into existing architectures with minimal effort. This setup converges in just 6 epochs while achieving state-of-the-art results.

## 5. Experiments & Results

### 5.1. Datasets & Training Configuration

We trained on the ASVspoof 2019 Logical Access (LA) set (Yamagishi et al., 2019), using the validation split for tuning. Evaluation covered ASVspoof 2021 (LA/DF) (Liu

et al., 2023) and the In-the-Wild corpus (Müller et al., 2022) to assess generalization. We handled class imbalance via weighted cross-entropy and applied RawBoost augmentation (Tak et al., 2022a) on 4s clips, consistent with prior work (Tak et al., 2022b; Xiao & Das, 2024; Rosello et al., 2023). Models were optimized using AdamW (lr $10^{-5} \rightarrow 10^{-8}$) on 4 NVIDIA L40 GPUs (effective batch 112). We report the best of three independent runs based on validation EER. Code, pretrained checkpoints, and an interactive demo are available at `https://github.com/idonithid/SONAR-Audio-DF-Detection`.

### 5.2. Results

**SONAR** achieves new state-of-the-art performance across DF, LA, and IN THE WILD (Table 1):

- **DF:** *SONAR-Full* and *SONAR-Finetune* reach **1.57%** and **1.45%**, surpassing all prior methods.

- **LA:** Both competitive performance (**1.20%–1.55%**) and are the strongest models evaluated under a single-run protocol.

- **In The Wild:** *SONAR-Full* and *SONAR-Finetune* set a new benchmark at **6.00%** and **5.43%**.

**Evaluation Protocol & Discussion.** To ensure a fair, deployment-realistic comparison, we evaluate SONAR under a strict *single-run* protocol, reporting the mean of three independent trainings without the checkpoint averaging or ensembling used by baselines (Xiao & Das, 2024). In this rigorous setting, SONAR achieves state-of-the-art performance on DeepFake and In-the-Wild benchmarks. The slight performance gap on the LA subset reflects this stricter regime, as SONAR prioritizes the robust, generalizable detection of frequency-dependent artifacts over the in-domain stability often inflated by weight averaging.

*Table 2.* **Ablation study (SONAR-Full).** Top: pooled EER (%) on DF, LA, and ITW sets under different architectural ablations. Bottom: robustness analysis of SONAR-Full (trained on the standard ASVspoof2019 dataset at 16 kHz) evaluated on the ITW test set due to its difficulty under different resampling and codec augmentations. Results are reported as probability shifts in the softmax outputs.

| Method / Augmentation | DF ↓ | LA↓ | ITW ↓ |
|---|---|---|---|
| *Architectural Ablations* | | | |
| SONAR-Full w/o RFE, JS | 2.54 | 2.93 | 8.91 |
| SONAR-Full w/o RFE | 2.40 | 2.48 | 8.44 |
| SONAR-Full w/o JS | 2.65 | 2.90 | 8.50 |
| SONAR-Full w/ non-learnable RFE (M=30) | 2.30 | 2.36 | 8.20 |
| SONAR-Full w/ $M{=}1$ SRM | 2.83 | 2.91 | 8.00 |
| SONAR-Full w/ $M{=}10$ SRM | 2.43 | 2.51 | 7.40 |
| SONAR-Full w/ Shared Encoder Weights | 2.50 | 2.65 | 8.40 |
| SONAR-Full w/ $\lambda_{JS}{=}0.5$ | 2.42 | 2.61 | 7.91 |
| SONAR-Full w/ $\lambda_{JS}{=}0.8$ | 1.90 | 1.78 | 7.02 |
| SONAR-Full w/ SRM and $\lambda_{JS}{=}1$ (best configuration) | **1.57** | **1.55** | **6.00** |
| *Robustness to Sampling Rates / Codecs (ITW only)* | | | |
| Resample → 44.1 kHz | | | $\lvert\Delta p\rvert = 0.03$ |
| Resample → 48 kHz | | | $\lvert\Delta p\rvert = 0.01$ |
| MP3 (64 kbps) | | | $\lvert\Delta p\rvert = 0.08$ |
| Opus (32 kbps) | | | $\lvert\Delta p\rvert = 0.08$ |
| Vorbis (q3) | | | $\lvert\Delta p\rvert = 0.04$ |

**Convergence speed.** SONAR also converges rapidly: while Tak et al. (Tak et al., 2022b) trained for 100 epochs, SONAR-Full stabilizes in 12 epochs, and SONAR-Finetune in only 4–6. Despite the additional branch, SONAR attains higher accuracy with nearly an order-of-magnitude faster training. This acceleration is driven by the Jensen–Shannon alignment loss (Eq. 10), which tightens low–high frequency coupling early in training and speeds up the separation of real and fake embeddings.

### 5.3. Ablation & Analysis

**Spectral Probing: Verifying the "Blind Spot"** To explicitly verify that SONAR overcomes spectral bias and generalizes to unseen domains, we conducted a probing experiment designed to isolate high-frequency discriminative cues. We took the models trained on ASVspoof 2019 and evaluated them on a filtered version of the **In-the-Wild** test set, retaining only frequencies above 4 kHz (High-Pass Cutoff). This setup removes the low-frequency semantic content that standard models typically overfit to, testing whether the models effectively learned generalizable high-frequency artifacts.

We extracted embeddings from both the Baseline and SONAR (Guided) encoders and trained a linear probe (Logistic Regression) to distinguish real from fake audio based solely on these high-frequency representations.

**Results.** As shown in Table 4 and Figure 5, the Baseline models struggle to classify the high-frequency signal, yielding high EERs of 32.51% ((Jung et al., 2022)) and 29.77% ((Xiao & Das, 2024)). This confirms our hypothesis: when

no emphasis is placed on high-frequency content during training, standard encoders suffer from spectral bias, effectively becoming "blind" to artifacts once low-frequency correlations are removed. In contrast, the SONAR-guided models achieve significantly lower error rates of 26.57% and 26.08% respectively. This demonstrates that our alignment objective successfully forces the encoder to capture generalizable high-frequency residuals, transforming the model's "blind spot" into a reliable discriminative signal even when semantic content is absent.

Table 2 validates the necessity of each SONAR component.

**Learnable vs. Fixed Filtering.** While fixed SRM filters provide only negligible gains over the baseline (8.44% → 8.20% EER), our learnable RFE achieves 6.00%. This confirms that static spectral heuristics are insufficient for diverse, in-the-wild attacks, whereas SONAR's data-driven filters successfully adapt to capture non-stationary artifact distributions.

**Necessity of Disentangled Encoders.** We found that sharing encoder weights across branches degraded performance to 8.40%, only improving slightly over the baseline (10.46%). This indicates that semantic content and high-frequency residuals possess fundamentally different distributions.

**Impact of Alignment.** The Jensen-Shannon alignment is equally critical; removing it causes sharp performance drops, particularly on the Logical Access partition. Furthermore, robustness tests (Table 2, bottom) confirm that these learned frequency-contrastive cues remain stable under common

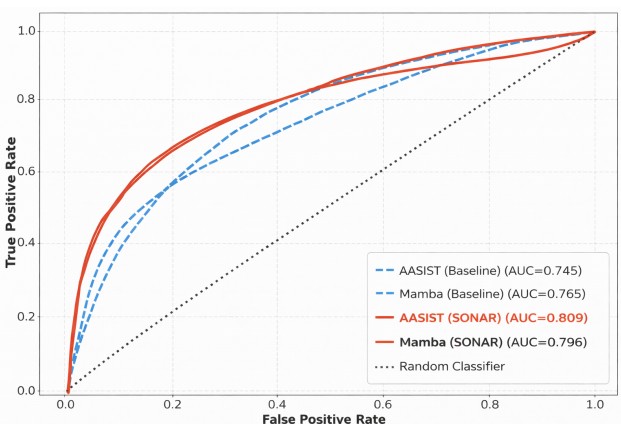

*Figure 5.* **Probing Spectral Sensitivity on In-the-Wild Data.** ROC curves for linear probes trained on High-Pass Filtered ($> 4$ kHz) audio embeddings from the In-the-Wild dataset. The **SONAR models (solid red/orange lines)** maintain high discriminative power in the high-frequency range (AUC $\approx 0.80$), whereas the **Baselines (dashed blue lines)** degrade significantly (AUC $\approx 0.75$). This performance gap confirms that baselines rely heavily on low-frequency correlations, while SONAR successfully captures generalizable high-frequency artifacts.

codec and resampling degradations.

**Sensitivity to bandwidth.** Because SONAR explicitly leverages high-frequency residuals, its accuracy degrades monotonically when the input bandwidth is artificially reduced from 16 to 4 kHz, a property we view as direct evidence that the model genuinely exploits HF synthesis artifacts rather than a weakness of the method (full sweep in Fig. 7, App. C). In practice, most audio captured *in the wild* retains $\geq 16$ kHz bandwidth (mobile capture, broadcast, streaming codecs); for narrow-band deployments, fine-tuning on bandwidth-matched data restores most of the accuracy.

**Robustness to neural codecs.** Beyond traditional codecs (MP3, Opus, Vorbis), we also assessed whether SONAR remains discriminative after re-encoding through a modern *neural* codec. We re-evaluated the pretrained SONAR-Full model zero-shot on the In-the-Wild test set after each clip is passed through the Descript Audio Codec (DAC, 16 kHz) (Kumar et al., 2023) encoder–decoder, without retraining. EER rises modestly from $6.00\%$ to $8.60\%$ and AUC from 0.977 to 0.953; crucially, the per-utterance score ranking is largely preserved (Spearman $\rho=0.944$, see Fig. 8 in App. C). This indicates that the learned HF–LF alignment captures cues that survive heavy neural re-synthesis, even though such codecs aggressively reshape the high-frequency band that other detectors depend on.

**Capacity, convergence, and real-time inference.** SONAR's gains are not due to increased capacity: a

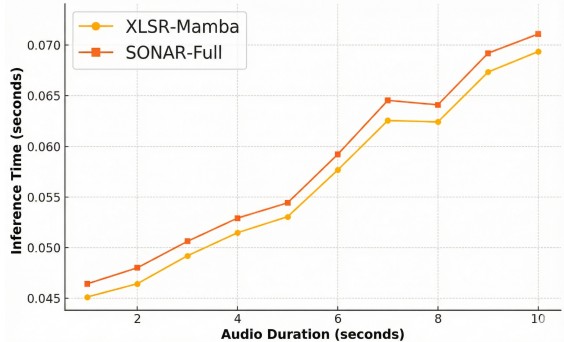

*Figure 6.* **Inference latency scales linearly with audio length.** We compare inference times (in seconds) for the XLSR-Mamba and SONAR-Full models across increasing audio durations from 1 to 10 seconds. SONAR introduces only a minimal overhead relative to XLSR-Mamba, while delivering improved detection performance (cf. Table 1).

*Table 3.* Parameter count, FLOPs (one 4-second @ 16 kHz forward), and inference latency.

| Model | Params | FLOPs | Latency (ms) |
|---|---|---|---|
| Single-encoder baseline | 317.84M | 150.41G | 29.0 |
| SONAR-Full (ours) | 641.53M | 303.92G | 57.3 |

dual-encoder model without RFE or JS alignment performs markedly worse on ITW ($8.91\%$) than full SONAR ($6.00\%$) despite identical parameter counts (Table 2). End-to-end measurements (Table 3) confirm that the dual-path design roughly doubles parameters and FLOPs relative to a single-encoder XLSR+AASIST baseline, while the RFE and cross-attention fusion contribute $< 1\%$ of either FLOPs respectively. Crucially, this analytical doubling *does not* translate to a doubling in wall-clock latency: because the two streams execute concurrently on GPU, single-batch inference for a 4 s clip at 16 kHz rises only slightly as shown in Fig. 6. SONAR therefore improves detection through better representation learning while remaining real-time feasible on a single commodity GPU.

**Embedding Analysis.** To assess latent separation, we evaluated SONAR-Full on the In-The-Wild set. Figure 4, show that the alignment loss preserves strong LF–HF coupling in real speech (cosine similarity near 1), while fakes collapse toward negative values ($\approx -0.2$).

## 6. Conclusion

**SONAR** reframes audio–deepfake detection as a *frequency-guided, contrastive* representation task. By splitting speech into complementary low- and high-frequency paths and regulating their latent divergence with a Jensen–Shannon loss, SONAR turns the generator's persistent "HF hole" into a decisive discriminative cue. Entirely data-driven, it requires no

*Table 4.* **High-Frequency Probing Results.** Performance of linear probes trained on pooled embeddings extracted from audio containing only high frequencies ($> 4$ kHz). The gap between Baseline and SONAR models indicates that SONAR encodes high-frequency artifacts ignored by baselines.

| Encoder | EER (%) ↓ | AUC ↑ | Acc@EER ↑ |
|---|---|---|---|
| XLSR-AASIST (Baseline) | 32.51 | 0.745 | 0.675 |
| **XLSR-AASIST (SONAR)** | **26.57** | **0.809** | **0.734** |
| XLSR-Mamba (Baseline) | 29.77 | 0.765 | 0.702 |
| **XLSR-Mamba (SONAR)** | **26.08** | **0.797** | **0.739** |

hand-crafted filters and offers a new lens for robust deepfake detection.

Empirically, SONAR-FULL and SONAR-FINETUNE achieve single-run **state-of-the-art** EERs with faster convergence than reported in strong baselines, while maintaining robustness to codecs and realistic bandwidth shifts.

**Key Breakthroughs.**

- **Spectral bias as a central generalization bottleneck.** We present strong empirical evidence that spectral bias plays a central role in the poor generalization of existing audio deepfake detectors. Probing experiments show that baseline models over-rely on low-frequency correlations and become effectively blind to high-frequency artifacts under distribution shift, while SONAR consistently captures and exploits these residual cues for robust detection.

- **Data-driven alignment of content and noise representations.** SONAR introduces a frequency-guided, data-driven alignment mechanism that couples semantic content with high-frequency residuals in latent space. Using learnable SRM filters and a Jensen–Shannon alignment loss, SONAR promotes coherent LF–HF structure for genuine speech and amplifies its systematic breakdown in deepfakes, elevating high-frequency residuals from auxiliary cues to a central supervisory signal.

- **SOTA accuracy with fast convergence.** SONAR achieves new single-run state-of-the-art EERs of 1.57% (DF), 6.00% (ITW), and 1.55% (LA) within 12 epochs, while SONAR-Finetune further improves performance to 1.45% / 5.43% (DF/ITW) and 1.20% (LA) in as few as 4 epochs.

## Impact Statement

This research advances application-driven machine learning by proposing a novel method for detecting AI-generated speech. By effectively identifying audio deepfakes and curbing malicious uses of voice-cloning models, it holds significant potential for positive social impact. A possible negative outcome is that publishing detector internals may help adversaries craft subtler manipulations that evade SONAR's frequency cues. We mitigate this by releasing code and benchmarks so the community can re-evaluate and re-train as new spoof methods emerge, rather than relying on a static detector. We urge the research community to minimize negative impacts while leveraging the positive contributions of this work.

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

# A. XLSR Architecture Overview

**XLSR** (Cross-Lingual Speech Representations) is a large-scale multilingual model based on the **Wav2Vec 2.0** architecture, trained on over 400,000 hours of speech across 128 languages. It is designed to learn universal speech representations that generalize well across languages and tasks. XLSR extends Wav2Vec 2.0 with larger model capacity and multilingual pretraining.

**Core Components**

- **Feature Encoder.** The input waveform $x \in \mathbb{R}^T$ is first passed through a series of temporal convolutional layers that output a latent representation:
$$z = \text{ConvEncoder}(x) \in \mathbb{R}^{T' \times d}$$
where $T' \ll T$ due to downsampling, and $d$ is the channel dimension.

- **Quantization Module (Pretraining Only).** A Gumbel-softmax quantizer maps $z$ to discrete latent codes $q(z)$ sampled from a learned codebook. This discrete representation is used as a target in contrastive learning. The quantization module is discarded after pretraining. The encoded sequence $z$ is fed into a multi-layer Transformer:
$$c = \text{Transformer}(z)$$
For XLSR 300M, the Transformer consists of 24 layers, each with hidden size 1024, 16 self-attention heads, and feed-forward networks of dimension 4096.

- **Contrastive Objective (Pretraining).** The model is trained to distinguish the true quantized target $q(z)$ from a set of distractors using a contrastive loss, encouraging the model to learn meaningful representations without labels.

**Downstream Usage**

After pretraining, the quantizer and contrastive heads are removed. The contextualized features $c$ are used as inputs to downstream tasks such as speech recognition, speaker verification, or deepfake detection. In our work, we extract $c$ either in frozen mode or via finetuning, and feed it into a task-specific classifier.

# B. AASIST Architecture Overview

**AASIST** (Audio Anti-Spoofing using Integrated Spectra-Temporal Modeling) is a deep learning model designed for detecting spoofed audio in speaker verification systems. It combines spectra-temporal modeling with attention-based mechanisms to robustly capture discriminative features between genuine and fake audio, particularly under real world conditions.

**Core Components**

- **Learnable Frontend:** The raw waveform $x \in \mathbb{R}^T$ is first passed through a 1D convolutional frontend that acts as a learnable filterbank:
$$x_{\text{spec}} = \text{Conv1D}(x)$$
This mimics handcrafted feature extraction (e.g., STFT or filterbanks) in a data-driven way and outputs time-frequency like representations.

- **Graph Attention Layer (GAT):** The core innovation of AASIST is to treat the spectro-temporal representation as a graph where each node corresponds to a time-frequency patch. A Graph Attention Network (GAT) models the structured relationships between these patches:
$$h_i' = \sum_{j \in \mathcal{N}(i)} \alpha_{ij} \mathbf{W} h_j$$
where $\alpha_{ij}$ are attention weights learned over neighbors $\mathcal{N}(i)$, and $\mathbf{W}$ is a shared linear transform.

- **Spectro-Temporal Blocks:** A series of convolutional blocks capture local patterns in both time and frequency domains. These are alternated with GAT layers to jointly model local and global context.

**Model EER (%) On In-The-wild Test Set With Different Sample Rates**

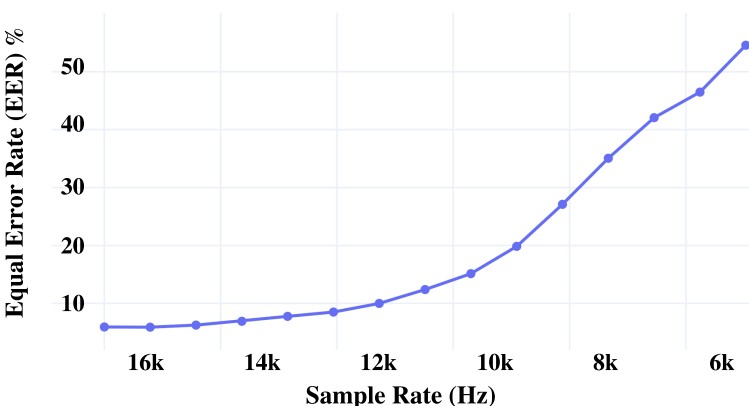

*Figure 7.* **Bandwidth sensitivity.** EER as a function of test-time sample rate. SONAR's dependence on HF cues is visible as a monotonic degradation when the input is downsampled, confirming that the model exploits exactly the high-frequency artifacts produced by neural synthesis.

- **Global Aggregation and Classification:** After the GAT and convolutional layers, the model aggregates features via global average pooling and passes them through fully connected layers for binary classification:

$$\hat{y} = \sigma(\text{MLP}(\text{GAP}(H)))$$

**Advantages**

- **Spectro-Temporal Awareness:** By combining CNNs and GATs, AASIST captures both fine-grained local patterns and long-range spectral dependencies.

- **Fully Learnable Pipeline:** From waveform to classification, the architecture is end-to-end trainable without handcrafted features.

- **Strong Benchmarks:** AASIST achieves state-of-the-art performance on ASVspoof 2019 and 2021 logical access (LA) and deepfake (DF) subsets, especially under noisy and real world conditions.

**Usage in Our Work**

We adopt AASIST as a strong baseline in our experiments on **SONAR-Full** model. Its ability to detect both TTS and VC-based attacks makes it a competitive model for evaluating deepfake detection methods.

## C. Robustness Analyses

We collect here the two robustness studies referenced in Sec. 5: bandwidth sensitivity (Fig. 7) and zero-shot resilience to a modern neural codec (Fig. 8).

**Bandwidth sensitivity.** Sweeping the test-set sample rate from 16 to $4\,\text{kHz}$ (stripping energy above the new Nyquist frequency at each step) causes EER to rise monotonically from a state-of-the-art $\sim 6\%$ at $16\,\text{kHz}$ toward random-guessing ($\sim 50\%$) at $4\,\text{kHz}$. We view this not as a weakness of the method but as direct empirical evidence that the model genuinely exploits HF synthesis artifacts—when those artifacts are removed by aggressive low-pass filtering, the discriminative signal disappears. In practice, most audio captured *in the wild* retains $\geq 16\,\text{kHz}$ bandwidth (mobile capture, broadcast, streaming codecs); for narrow-band deployments, fine-tuning on bandwidth-matched data restores most of the accuracy.

**Neural codec robustness.** Beyond traditional codecs (MP3, Opus, Vorbis), we additionally re-evaluated the pretrained SONAR-Full model zero-shot on the In-the-Wild test set after each clip was passed through the Descript Audio Codec (Kumar

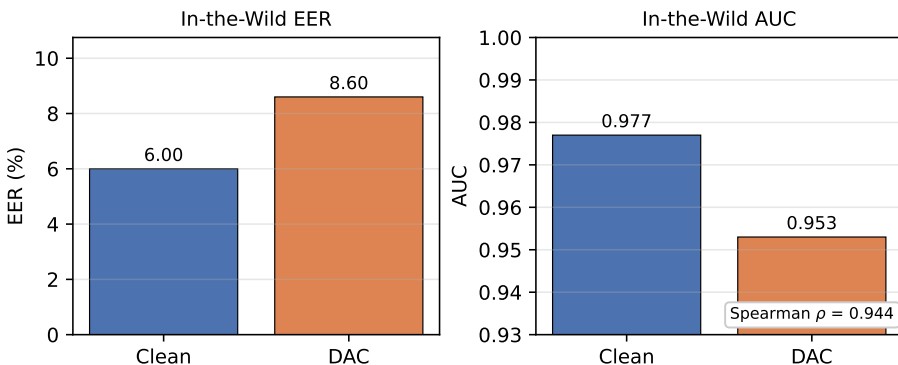

*Figure 8.* **Neural codec robustness.** Zero-shot evaluation of pretrained SONAR-Full on In-the-Wild before/after Descript Audio Codec encode–decode at 16 kHz. Score ranking is preserved (Spearman $\rho = 0.944$).

et al., 2023) (DAC, 16 kHz) encoder–decoder, with no retraining. EER rises modestly from $6.00\%$ to $8.60\%$ and AUC from $0.977$ to $0.953$, while per-utterance score ranking is largely preserved (Spearman $\rho=0.944$). The learned HF–LF alignment therefore captures cues that survive heavy neural re-synthesis, even though such codecs aggressively reshape the high-frequency band that other detectors depend on.

## D. Limitations

The bandwidth-sensitivity analysis above (App. C) doubles as a limitations check: while $\geq 16$ kHz capture is the norm in the wild, the detector becomes less robust when real-world pipelines apply aggressive low-pass filtering or resampling. Practitioners should therefore (i) preserve as high a capture sample-rate as feasible, or (ii) retrain / fine-tune the model on data reflecting the target bandwidth and compression conditions.

**Model size and compute.** Although the dual-path design roughly doubles the parameter count to 650 M (with XLSR large), it remains feasible to train for 12 epochs on a single L40 GPU standard for real world remote server deployments. We leave further optimization via parameter sharing and pruning for future work.

**Modality scope.** Experiments are confined to audio. While the frequency-guided principle is generic, porting SONAR to images or video will require modality-specific high-pass filtering and fusion schemes, which we have not yet explored.

**Dataset coverage.** Evaluation spans ASVspoof 2021 (LA/DF) and the Müller *in-the-wild* corpus, although these are the academic benchmarks for spoofing detection, unseen spoof mechanisms or languages may still degrade performance.

**False positives/negatives.** Like any detector, SONAR can misclassify highly compressed real speech or exceptionally well-crafted fakes, which could erode user trust. Threshold calibration for different deployment domains remains an open question.

## E. Licensing of Third-Party Assets

All third-party assets used in this work, including pretrained models (Table 5) and datasets (Table 6), are listed below along with their licenses and source URLs. All components comply with their respective open-source or research-use licenses.

## F. Reproducibility

We build directly on the publicly released AASIST and XLS-R reference implementations, adopting the CUDA-optimised training framework of Tak et al. All experiments were run end-to-end on a single NVIDIA L40 (48 GB) GPU under PyTorch 2.2 with CUDA 12.2. The complete source code, Hydra configs, pretrained checkpoints, and the shell scripts used to reproduce every table and figure accompany this paper in `https://github.com/idonithid/SONAR-Audio-DF-Detection`. Broader societal considerations are discussed in the main-body Impact Statement.

| Model | License | URL |
|---|---|---|
| XLSR (fairseq) | MIT | https://github.com/facebookresearch/fairseq |
| XLSR-Mamba | MIT | https://github.com/swagshaw/XLSR-Mamba |
| AASIST | MIT | https://github.com/clovaai/aasist |

*Table 5.* Licenses for pretrained models.

| Dataset | License | URL / Terms |
|---|---|---|
| In The Wild | Apache 2.0 | https://deepfake-total.com/in_the_wild |
| ASVspoof 2019 (LA/DF) | ODC-By v1.0 | https://datashare.ed.ac.uk/handle/10283/3336 |
| ASVspoof 2021 (LA/DF) | ODC-By v1.0 | https://doi.org/10.5281/zenodo.4837263 |

*Table 6.* Licenses for datasets.

# G. Computational Cost

We analyzed the additional cost of SONAR relative to a single-stream XLSR baseline. The extra components are: (i) the Rich Feature Extractor (RFE), (ii) a second encoder branch, and (iii) a cross-attention fusion.

**RFE and fusion are negligible.** For a 4 s clip at 16 kHz with $M{=}10$ filters, the RFE adds only $\sim$8M FLOPs ($< 0.01$G), and the cross-attention adds $\sim$0.16G FLOPs. Both are $< 1\%$ of a single XLSR pass.

**Encoders dominate.** SONAR-Full essentially doubles the encoder cost, giving $\approx 2\times$ the parameters and FLOPs of XLSR. However, since the two streams run concurrently on GPU, the measured wall-clock latency increases by only 15–25% (Fig. 6), not 100%.

**End-to-end measurement.** Table 3 reports parameter count, multiply–accumulate FLOPs (one forward on a 4 s @ 16 kHz clip, computed with `calflops`), and single-batch inference latency on an NVIDIA Quadro RTX 8000 for the single-encoder XLSR+AASIST baseline and SONAR-Full. The measurements confirm the analytical $\approx 2\times$ estimate above: parameters scale from $317.84$ M to $641.53$ M (factor 2.02) and FLOPs from $150.4$ G to $303.9$ G (factor 2.02), while wall-clock latency rises only from $29$ ms to $57$ ms.

