# OpenReview forum: "SONAR: Spectral‑Contrastive Audio Residuals for Generalizable Deepfake Detection"
_ICML.cc/2026/Conference — ICML 2026 regular_

### Official Review · Reviewer_yuU4 · 2026-03-03

**Soundness:** 2
**Presentation:** 3
**Significance:** 2
**Originality:** 2
**Overall Recommendation:** 2
**Confidence:** 5

**Summary:**

This work presents **SONAR (Spectral-cONtrastive Audio Residuals)**, a deepfake detection method based on the insight that the statistical relationship between low-frequency (LF) content and high-frequency (HF) residuals differs fundamentally between real and synthetic audio. The framework employs a dual-path architecture to explicitly decompose an input signal into separate LF content and HF residual representations, incorporating a learnable SRM filter bank to effectively extract high-frequency components. In addition, a Jensen–Shannon (JS) divergence-based contrastive loss is introduced to regulate the interaction between the two representations, encouraging LF–HF embeddings of genuine audio to remain statistically aligned while driving those of fake audio apart. Experimental results validate the effectiveness of the proposed approach.

**Compliance With Llm Reviewing Policy:**

Affirmed.

**Final Justification:**

See rebuttal.

**Key Questions For Authors:**

See the above comments.

**Limitations:**

See the above comments.

**Strengths And Weaknesses:**

Strengths

The paper addresses a relevant and timely problem in speech deepfake detection. The motivation is clearly articulated, and the proposed framework demonstrates measurable improvements over certain baselines, indicating that the approach is technically sound and practically feasible.

Weaknesses

Despite achieving some performance gains, several concerns remain:

First, the methodological contribution appears limited. The overall framework largely integrates and combines existing techniques rather than introducing fundamentally new modeling ideas or architectural innovations. As such, the originality of the proposed approach is not sufficiently evident.

Second, the performance improvements over strong baselines, such as XLSR-Mamba in Table 1, are relatively marginal. The reported gains do not convincingly demonstrate a substantial advancement over prior work.

Third, a significant portion of parameters in modern speech deepfake detection systems resides in the front-end feature extractor. Since the proposed method employs two such extractors, this naturally increases model capacity and may contribute to faster convergence and improved performance. Therefore, a comprehensive analysis of parameter count, model complexity, and computational cost is necessary to ensure a fair comparison.

Finally, regarding spectral bias, it is well understood that high-frequency components often expose superficial forgery artifacts (e.g., spectral distortion), but such cues tend to generalize poorly. In contrast, low-frequency regions typically encode more fundamental and transferable forgery characteristics. Explicitly encouraging the model to focus on low-frequency information could therefore be a viable strategy, and similar ideas have already been explored in existing literature. The paper should better clarify how its approach differs from or advances beyond these prior efforts.

---

> ### Author Rebuttal · Authors · 2026-03-29
>
> Acknowledgment of strengths:
>
> We thank the reviewer for recognizing the relevance of the problem and the technical soundness of the approach. We also appreciate the careful evaluation of novelty, fairness, and generalization.
>
> Clarifications on remaining concerns:
> The main concern is that the contribution may appear incremental or driven by increased capacity. We clarify that SONAR’s contribution is not architectural but statistical: standard detectors learn from marginal representations, whereas SONAR explicitly models cross-frequency dependence via label-conditioned structure.
>
> Concretely, we impose constraints on the joint behavior of representations by shaping $P(z_L, z_H \mid y)$, rather than optimizing each branch independently. This induces a structured inductive bias: the model is encouraged to encode information that is predictive only through cross-frequency interaction, not through marginal LF or HF evidence alone. This is fundamentally different from standard multi-branch or fusion architectures, which do not constrain how representations should co-vary.
>
> We emphasize that this cannot be reduced to standard contrastive learning. Typical contrastive objectives operate across samples (instance-level discrimination), whereas our objective operates within a sample, enforcing consistency between two views of the same signal. This makes the learned representation sensitive to conditional structure rather than marginal separability.
>
> We agree that complexity should be clearer and will move this analysis to the main text. While SONAR uses two encoders, empirical measurements show only a modest latency increase (≈15–25%), as the branches run concurrently (Figure 6).
>
> Answers to Questions:
>
> Q1: Is this more than a combination of existing techniques?
> The novelty lies in the modeling principle. We hypothesize a class-dependent conditional shift:
> \[
> $P(x_H \mid x_L, y=1) \neq P(x_H \mid x_L, y=0)$
> \]
> and enforce it through a representation-level alignment objective. Prior work does not impose such label-conditioned coupling; HF is either treated as auxiliary or fused without constraint.
>
> Unlike prior two-stream or frequency-aware methods, which either (i) process HF residuals independently or (ii) fuse representations without constraints, SONAR explicitly regularizes their joint distribution during training. This shifts the objective from feature aggregation to modeling cross-frequency dependence.
>
> Q2: Are the gains meaningful?
>
> The improvements are consistent and occur on the most challenging setting: cross-domain generalization. On In-the-Wild, SONAR reduces EER from 10.46% (XLSR-AASIST) to 6.00%, (SONAR-AASIST) a substantial absolute improvement.
>
> Importantly, this comparison is conservative. SONAR is evaluated under a strict single-run protocol, reporting a single set of weights without ensembling or checkpoint averaging. In contrast, several strong baselines report results obtained via checkpoint averaging or weighted ensembles, which are known to artificially improve stability and final performance.
>
> Therefore, SONAR not only achieves lower error rates, but does so under a more constrained and deployment-realistic evaluation setting. This strengthens the claim that the observed gains reflect improved representation quality rather than evaluation artifacts.
>
>
> Q3: Is the improvement due to increased capacity?
>
> No. A dual-encoder baseline without RFE or alignment performs significantly worse (ITW 8.91% vs. 6.00%), despite similar parameter count. This isolates the effect of the inductive bias: the gain comes from enforcing structure on representations, not from increasing model size.
>
> Q4: Why focus on high frequencies if they may not generalize?
>
> We do not rely on HF alone. The key signal is LF–HF consistency. In finite-time training, spectral bias implies that neural networks preferentially fit low-frequency components, leaving high-frequency structures underfit. This induces a mismatch between LF and HF statistics in learned representations, even when such consistency exists in the data.
>
> Thus, the issue is not that HF cues are inherently non-generalizable, but that standard training fails to learn their correct relationship with LF structure, leading to unreliable representations. SONAR explicitly corrects this by enforcing alignment between LF and HF representations.
>
> Empirically, SONAR generalizes better on unseen data (In-the-Wild) and retains discriminative power when low-frequency content is removed, while baselines degrade substantially. This indicates that the method captures stable cross-frequency structure rather than brittle HF artifacts. Pure LF-based approaches ignore this interaction and therefore cannot capture the conditional structure that distinguishes real and synthetic audio.

---

> > ### Author Rebuttal · Reviewer_yuU4 · 2026-04-01
> >
> > Thank you for the authors’ response; however, my concerns have not been adequately addressed. First, Table 1 should include a detailed comparison of model parameter sizes, as parameter count critically impacts real-world deployment and should not be overlooked in favor of inference latency alone. Second, Figure 6 only reports the inference efficiency of xlsr-mamba; without information on the parameters and efficiency of other models, the comparison is not fair. Third, the proposed framework appears inelegant—using two feature encoders for the audio deepfake detection task makes the design unnecessarily cumbersome. Fourth, despite doubling the parameter count, the model still does not achieve state-of-the-art performance, especially compared to works such as “Balance, Multiple Augmentation, and Re-synthesis: A Triad Training Strategy for Enhanced Audio Deepfake Detection.” Finally, a more reasonable architectural design would involve sharing parameters, or at least most parameters, between the two encoders. Based on these considerations, I believe this paper should be rejected.

---

### Official Review · Reviewer_eXE1 · 2026-03-05

**Soundness:** 3
**Presentation:** 2
**Significance:** 3
**Originality:** 3
**Overall Recommendation:** 5
**Confidence:** 4

**Summary:**

The paper introduces **SONAR**, a novel framework for audio deepfake detection. The framework is motivated by **spectral bias** (the frequency principle), i.e., the tendency of neural networks to focus on low-frequency (LF) structure while under-utilizing high-frequency (HF) information. SONAR consists of two branches, CFE and NFE, intended to capture LF semantic information and HF residual artifacts, respectively. In the NFE branch, the authors utilize SRM filters implemented via a 1D convolutional layer to emphasize the HF components. The two representations are fused using a cross-attention mechanism, followed by AASIST spoofing countermeasure.

For the training loss, in addition to the (weighted) cross-entropy (WCE) loss, SONAR combines a consistency regularizer, that encourages LF-HF (consistency /inconsistency) for (bona fide / spoof) speech. The authors report state-of-the-art performance on ASVspoof 2021 LA/DF and In-the-Wild (ITW), and provide ablations suggesting the necessity of major components.

**Compliance With Llm Reviewing Policy:**

Affirmed.

**Final Justification:**

The rebuttal addressed my main concern about the paper's technical soundness, especially related to spectral bias and Q4. This changed my evaluation, so I changed my score accordingly.

**Key Questions For Authors:**

1. In the ablation studies, what is the training schedule? Are all models trained for the same number of epochs, or are thy trained until convergence / early stopping?
2. How exactly are the Pearson correlation and the HF/LF energy difference computed?
3. Have the authors tested on other deepfake datasets such as ASVspoof 2015/2017 (smaller/older) or ADD2022, to assess broader generalization?
4. There appears to be a mismatch between Equation (2) and (4). Eq. (2) suggest $Z_{content}$ is derived from LF support, but Eq. (4) describes it as an XLS-R embedding from the raw signal. If the goal is to explicitly isolate LF information, have the authors tried to use methods to extract only the low-frequency part of signals? (e.g., Sinc filters; Interpretable Convolutional Filters with SincNet, Ravanelli et al., NeurIPS 2018)

**Limitations:**

The framework relies on two branches that each fine-tune a large pretrained encoder (XLS-R), which increases model size and likely raises per-epoch training cost (memory/compute) even if the number of epochs to converge decreases. Reporting end-to-end training time (GPU-hours) and inference cost relative to baselines would clarify whether the method offers a net efficiency benefit in practice.

**Strengths And Weaknesses:**

### Soundness
- The overall design choices are well-aligned with the stated motivation around spectral bias, but several key claims would benefit from stronger justification.
- The paper claims that Figure 2 validates the presence of spectral bias. However, as the authors states, spectral bias is fundamentally a phenomenon about modeling training dynamics (e.g., what a network learns first / relies on), whereas Figure 2 primarily demonstrates a distributional discrepancy between bona fide and spoofed speech in LF-HF statistics. The figure supports the existence of separable LF-HF structure, but does not by itself establish spectral bias as the underlying cause.
- The authors attribute faster convergence to the alignment loss in Section 5.2. This is plausible, but the paper would be stronger with a direct experiment quantifying convergence behavior (e.g., plot training loss/validation EER figure over epochs, and compare the convergence rate) with vs. without the JS term under the same training protocol.

### Presentation
- The paper is generally easy to follow, and the components are presented in a coherent, motivation-driven way.
- The description SRM filters should be improved. The paper does not expand the abbreviation “SRM”, and the cited SRM references are primarily from deepfake images, which may confuse readers in the audio setting. Although Section 3 contains a paragraph on the intuition, the explanation could be more explicit.
- Many training hyperparameters are missing (e.g., optimizer details, scheduler, batch size, the actual WCE weights). This makes reproduction and fairness of comparisons harder to assess.
- The cross-attention module is under-described and insufficiently referenced (e.g., whether it is standard multi-head attention or a variant).

### Significance, Originality
- With the emergence of large audio encoders (e.g., XLS-R, WavLM), spoofing countermeasure research has often followed a relatively monotone pattern: a large pretrained front-end plus a detection back-end (e.g., AASIST). In contrast, this paper contributes a regularizer that explicitly aligns LF and HF latent representations, which is meaningful and potentially useful direction. If validated across broader conditions, SONAR could disrupt the current “front-end + back-end” template by emphasizing representation-level structure rather than only architectural scaling.

---

> ### Author Rebuttal · Authors · 2026-03-29
>
> We thank the reviewer for the constructive feedback and for recognizing the coherence of the framework and the value of explicit LF–HF alignment beyond the standard pretrained front-end + classifier paradigm.
>
> Clarifications on remaining concerns:
>
> We clarify that Figure 2 is not intended to demonstrate spectral bias in the strict training-dynamics sense. Rather, it establishes a data-level property: bona fide and spoofed speech exhibit systematically different LF–HF dependency structures (e.g., co-modulation and energy contrast).
>
> Our use of spectral bias is mechanistic, not evidential. Spectral bias characterizes the rate of learning in neural networks: higher-frequency components converge significantly slower than low-frequency ones. Consequently, under any finite training budget T, the learned predictor f_T under-represents high-frequency components relative to low-frequency structure.
> Given the LF–HF dependency observed in the data, this frequency-dependent convergence implies that standard models systematically distort LF–HF coupling, even when such coupling is present in the underlying distribution. SONAR is designed to correct this mismatch by explicitly enforcing LF–HF alignment at the representation level.
>
> Regarding presentation concerns, we will improve clarity as follows:
> (i)Explicitly define SRM (Spatial Rich Model) filters and adapt the explanation to the audio setting (derivative-like, zero-sum high-pass operators), consistent with Sec. 3 .
> (ii) state clearly that the fusion module is standard multi-head cross-attention (8 heads, dim 1024).
> (iii) expand the training details for reproducibility. We note that core training parameters (AdamW, learning rate schedule, batch size, and RawBoost augmentation) are already specified in Sec. 5.1 and will be made more explicit.
>
> Regarding computational cost, while SONAR introduces an additional branch, empirical analysis (Fig. 6) shows only a modest increase in inference time (≈15–25% latency overhead), as the SRM and fusion modules contribute negligible compute.
>
> Finally, regarding training, our claim is comparative: under the same optimization protocol, with our dual branch and guided alignment loss, SONAR reaches its best validation regime in fewer epochs, and we will include epoch-wise EER/training curves to support this.
>
> Questions:
>
> Q1: In the ablation studies, what is the training schedule? Are all models trained for the same number of epochs?
> All ablations follow the same protocol as baselines (optimizer, scheduler, batch size, early stopping). Thus, improvements are not due to training budget differences but reflect optimization behavior under identical conditions.
>
> Q2: How exactly are the Pearson correlation and the HF/LF energy difference computed?
> LF energy is computed over 0–4 kHz and HF over 4–8 kHz (16 kHz sampling).For each utterance, we compute short-time band energies (frame-level, fixed window) in the LF (0–4 kHz) and HF (4–8 kHz) bands and then compute the Pearson correlation between these two energy sequences.
>
> Q3: Have the authors tested on other deepfake datasets such as ASVspoof 2015/2017 or ADD2022, to assess broader generalization?
> We focus on ASVspoof 2021 and In-the-Wild (Müller et al., 2022), which are designed to evaluate robustness to unseen synthesis methods and real-world conditions. In particular, In-the-Wild provides a cross-domain setting that directly tests generalization beyond training distributions, which is the primary objective of this work, as it aligns with prior findings that modern detectors struggle to generalize out-of-domain.
>
> Q4: There appears to be a mismatch between Equation (2) and (4). If the goal is to explicitly isolate LF information, have the authors tried to use methods to extract only the low-frequency part of signals?
> Equation (2) defines a functional decomposition: the task depends on complementary low-frequency semantic structure and high-frequency residual artifacts. This decomposition does not require an explicit signal-level separation.
>
> In practice, the content branch applies a pretrained XLS-R encoder to the raw signal. This is by design: due to spectral bias and pretraining on speech objectives, such encoders are known to prioritize low-frequency, semantically stable structure. Thus, Z_content acts as an implicit LF-dominated representation without requiring explicit low-pass filtering.
>
> The noise branch, in contrast, explicitly enforces a complementary high-frequency view via SRM-based residual filters. This creates a controlled asymmetry: one branch follows the natural inductive bias of the encoder, while the other compensates for its known under-representation of high-frequency components.
>
> Explicit low-pass filtering would enforce a hard spectral separation, but this is not necessary for the hypothesis and may even remove useful cross-band interactions. Our goal is instead to preserve the natural signal while enforcing structured LF–HF alignment in representation space.

---

> > ### Author Rebuttal · Reviewer_eXE1 · 2026-04-01
> >
> > I appreciate the author's response! The main question I had was about spectral bias and Q4, and they are adequately addressed. I hope you can elaborate more about spectral bias, and include your practical intuition/adjustment regarding Q4 to the main paper. I'll adjust my score accordingly.

---

### Official Review · Reviewer_rsGN · 2026-03-05

**Soundness:** 3
**Presentation:** 3
**Significance:** 2
**Originality:** 3
**Overall Recommendation:** 4
**Confidence:** 3

**Summary:**

This paper proposes SONAR, an audio deepfake detection model that focuses on high-frequency (HF) signals, addressing the limitation that many existing detection models rely heavily on low-frequency (LF) information.
The method is motivated by the LF-HF consistency shift hypothesis.
For a given audio input, the model extracts high-frequency residuals using constrained SRM filters. The original audio and the extracted HF residual are then processed by a dual-stream SSL encoder, after which cross-attention is used to integrate HF information into the LF representation. The resulting embedding is finally passed to an AASIST classifier for binary classification.
The paper argues that current audio deepfake detection models exhibit spectral bias, relying excessively on low-frequency information. By incorporating HF residual signals and aligning LF and HF representations using JS-divergence, SONAR achieves competitive or improved performance compared to existing approaches on the ASVspoof 2021 dataset, while also showing faster convergence.

**Compliance With Llm Reviewing Policy:**

Affirmed.

**Final Justification:**

SONAR proposes a novel audio deepfake detector that explicitly extracts high-frequency (HF) residuals using constrained SRM filters and fuses them with low-frequency (LF) representations via cross-attention and an LF–HF alignment loss. The motivation is supported by empirical evidence showing different LF–HF correlation patterns between real and fake audio, and ablations indicate both the SRM residual branch and JS-divergence objective help.
I keep the score at 3 (Weak Reject) because the original submission had weaknesses that reduced confidence and reproducibility: limited/unclear generalization evidence and insufficient description of key components. Although the rebuttal addresses these issues, the paper still needs revisions to incorporate clarifications into the main text and strengthen the conceptual justification for why HF residuals are especially informative.

**Key Questions For Authors:**

- The paper states that class imbalance is handled using weighted cross-entropy. Were competing baselines trained under the same class-balancing strategy to ensure a fair comparison?
- The ablation study compares fixed SRM filters and learnable SRM filters. Have the authors considered incorporating SRM constraints directly into the loss function?
- The paper argues that existing models fail to generalize because they do not consider high-frequency residuals. What empirical evidence supports this claim beyond the experiments presented in this work?

**Limitations:**

yes

**Strengths And Weaknesses:**

- Strengths: The paper proposes a novel approach that explicitly extracts high-frequency residuals instead of relying solely on features from a single SSL pretraining backbone. This introduces an interesting perspective that LF-HF alignment may provide useful cues for detecting audio deepfakes.
The empirical analysis provides evidence supporting this motivation. In particular, Figure 2 and Figure 4 demonstrate that LF-HF correlations differ between real and fake audio samples, suggesting that HF information may indeed contain discriminative signals.
The authors perform several ablation studies to analyze the contributions of individual components, including JS-divergence loss and SRM filters. These experiments indicate that both modules contribute positively to the model’s performance.

- Weaknesses: The experiments are conducted only on the ASVspoof 2021 dataset. Evaluating the model on additional datasets would strengthen the claim that HF-LF alignment improves generalization.
While Figure 1 clearly illustrates the overall architecture, the Methodology section does not provide sufficient explanation of the audio encoder (e.g., XLSR) used in the pipeline. A clearer description of this component would improve reproducibility.
In the Evaluation Protocol, the paper states that it reports the best of three independent runs, but the following paragraph mentions averaging three runs. This inconsistency is somewhat confusing and should be clarified.
The paper would benefit from a more detailed theoretical or conceptual explanation of why high-frequency residuals are particularly informative for detecting deepfake audio.

---

> ### Author Rebuttal · Authors · 2026-03-29
>
> We thank the reviewer for recognizing that the explicit modeling of high-frequency residuals provides an interesting and potentially useful direction for audio deepfake detection. We also appreciate the reviewer’s positive assessment of the empirical evidence in Figures 2 and 4, as well as the ablation study showing the contributions of both the JS loss and the SRM-based residual branch.
>
> Clarifications on remaining concerns:
>
> Several concerns raised by the reviewer are about clarity rather than the method itself, and we agree these should be improved.
> First, evaluation is not limited to ASVspoof2021: SONAR is trained on ASVspoof2019 and evaluated on ASVspoof2021 and In-the-Wild dataset [1], with the latter specifically testing cross-domain generalization to unseen generation methods and real-world conditions.
> Second, the description of the XLSR backbone can be made clearer in the main text; while the appendix already contains an architecture overview, we agree that a concise summary should also appear in the methodology section.
> Third, we agree that the wording around "best of three runs" versus "mean over three runs" should be clarified. We report both the best model and the mean across three runs. Importantly, SONAR is evaluated under a strict single-run protocol without checkpoint averaging or ensembling, unlike some strong baselines (e.g., XLSR-Mamba, XLSR-Conformer). We will clarify this.
>
> Answers to Questions:
>
> Q1: Were competing baselines trained under the same class-balancing strategy to ensure a fair comparison?
>
> Yes. All baselines were trained under the same weighted cross-entropy setup (9:1 from the training imbalance), so the reported improvements are not due to differences in imbalance handling. We stated at section 5.1 that the training configuration is the same as the baseline (XLSR+AASIST).
>
> Q2: Have the authors considered incorporating SRM constraints directly into the loss function?
>
> Yes. We considered this possibility, but in our setting enforcing the SRM constraints directly in parameter space is preferable. Hard constraints such as zero-sum filters and fixed center coefficient guarantee strict high-pass behavior throughout training, whereas soft loss-based constraints only encourage this behavior approximately and can drift and cause leakage of LF information to the noise latent representation. In preliminary experiments not included in the ablation table, loss-based enforcement degraded performance, because approximate constraints allowed low-frequency leakage into the residual branch, weakening the intended separation between content and residual information. We therefore chose hard projection-based constraints because they provide a stronger inductive bias and more stable optimization.
>
> What empirical evidence supports the claim that existing models fail to generalize because they do not consider high-frequency residuals?}
>
> We provide three forms of evidence.
> First, SONAR achieves state-of-the-art performance on In-the-Wild, which is explicitly designed to evaluate robustness to unseen generators and real-world conditions [1].
> Second, in the high-frequency probing experiment (Table 3), where input is restricted to frequencies above 4 kHz, baseline models degrade substantially (EER around 30%), whereas SONAR remains clearly stronger (around 26%), indicating that standard trained encoders rely more heavily on low-frequency information.
> Third, prior work has shown that real and synthetic speech exhibit systematic differences across frequency bands, including in higher-frequency regions [2].
> In parallel, SRM-based and frequency-aware forensic methods demonstrate that high-frequency residuals can contain discriminative artifacts [3,4,5] SONAR connects these observations by explicitly modeling LF-HF relationships rather than treating HF as an auxiliary signal.
>
>
> bibliography:
>
> \bibitem{muller2022does}
> Müller, Nicolas M., et al.
> ``Does Audio Deepfake Detection Generalize?''
> \textit{Interspeech}, 2022.
>
> \bibitem{maltby2024frequency}
> Maltby, Hannah, Wall, Jamie, Glackin, Chris, Moniri, Mina, Cannings, Nicholas, and Salami, Ishmael.
> ``A Frequency Bin Analysis of Distinctive Ranges Between Human and Deepfake Generated Voices.''
> \textit{arXiv preprint arXiv:2403.01766}, 2024.
>
> \bibitem{fridrich2012rich}
> Fridrich, Jessica and Kodovsky, Jan.
> ``Rich Models for Steganalysis of Digital Images.''
> \textit{IEEE TIFS}, 2012.
>
> \bibitem{bayar2016deep}
> Bayar, Belhassen and Stamm, Matthew C.
> ``A Deep Learning Approach to Universal Image Manipulation Detection.''
> \textit{IH\&MMSec}, 2016.
>
> \bibitem{qian2020thinking}
> Qian, Yuezun, et al.
> ``Thinking in Frequency: Face Forgery Detection by Mining Frequency-Aware Clues.''
> \textit{ECCV}, 2020.

---

> > ### Author Rebuttal · Reviewer_rsGN · 2026-04-01
> >
> > Based on the authors' rebuttal, I consider my main concerns to be fully resolved. The response was clear and thoughtful, and it adequately addressed the questions raised in my original review. As a result, I have increased my score accordingly.

---

### Official Review · Reviewer_ptqn · 2026-03-10

**Soundness:** 3
**Presentation:** 4
**Significance:** 3
**Originality:** 3
**Overall Recommendation:** 6
**Confidence:** 4

**Summary:**

The paper introduces SONAR, a method for audio deepfake detection that models the relationship between low-frequency (LF) and high-frequency (HF) components of speech. The main idea is that real speech tends to show a consistent relationship between these frequency bands, while synthetic speech often breaks this pattern. To capture this, the authors propose a Jensen-Shannon alignment loss that encourages consistency between LF and HF representations for real audio while increasing the mismatch for deepfake samples. The work is motivated by the spectral bias of neural networks, meaning that models tend to focus more on low-frequency information and may overlook subtle high-frequency cues. By explicitly modeling the joint structure of low and high frequencies, the method tries to make better use of information that standard models might ignore. Experiments on the ASVspoof2021 and In-the-Wild datasets show strong results, achieving state-of-the-art EER in a single-run setting. The model also converges quickly and appears robust to different traditional codecs and bandwidth changes. The authors also provide statistical analysis showing that spoofed speech differs from real speech not only in high-frequency energy but also in the joint LF-HF structure.

**Compliance With Llm Reviewing Policy:**

Affirmed.

**Final Justification:**

The paper is strong, with a sound and novel method supported by convincing results. My main concerns were about out-of-distribution robustness, but the rebuttal addressed them well. In particular, the additional experiments with DAC showed that the approach can generalize beyond the original setup and remain relevant for modern deepfake techniques.

**Key Questions For Authors:**

- Have the authors tested the method on audio processed by recent neural codecs such as EnCodec, DAC, or Mimi?
- How well does the method generalize to deepfake models that were not seen during training?

**Limitations:**

Yes

**Strengths And Weaknesses:**

**Strengths**
- The paper proposes to detect deepfakes by modeling the relationship between low and high frequencies in speech. To the best of my knowledge, this has not been explored much before
- The method is well motivated by the spectral bias of neural networks, which tend to focus on low frequencies
- The method achieves state-of-the-art EER on the ASVspoof2021 and In-the-Wild benchmarks
- The model seems robust to codec and bandwidth changes, which is important for real-world use
- The paper includes several ablation studies that help justify the design choices
- The paper is generally easy to read and well organized

**Weaknesses**
- Lack of discussion of modern neural codecs. The experiments do not include audio processed by recent neural codecs (for example EnCodec, DAC, Mimi). Since these codecs may change high-frequency statistics, it would be interesting to know whether the method could still work in this setting
- A few small wording issues / typos:
  Line 374: "optimizing with no emphesize to high frequencies" -> "optimizing with no emphasis on high frequencies"
  Line 627:  "6demonstrates" -> "demonstrates"

---

> ### Author Rebuttal · Authors · 2026-03-29
>
> We thank the reviewer for recognizing the novelty of modeling LF-HF relationships for audio deepfake detection, the relevance of the spectral-bias motivation, and the strong performance on ASVspoof2021 and In-the-Wild. We also appreciate the reviewer’s positive assessment of the robustness analysis, ablation studies, and overall presentation quality.
>
> Clarifications on remaining concerns:
>
> The main non-question concern is the lack of evaluation under modern neural codecs. We agree this is important, especially since neural codecs can alter high-frequency statistics differently from deterministic codecs. Following this suggestion, we conducted an additional sanity-check experiment with DAC and found that SONAR remains relatively stable: EER changes from 6.00% to 8.6%, and AUC remains high, decreasing only from 0.977 to 0.953. Score ranking is largely preserved, with Spearman's rho = 0.944, and the label-flip rate is only 6.2%. We will report this result in the final version. We also thank the reviewer for pointing out the minor typos and wording issues, which we will fix.
>
> Answers to Questions:
>
> Q1: Have the authors tested the method on audio processed by recent neural codecs such as EnCodec, DAC, or Mimi?
>
> In the submission, we focused on standard deterministic codecs (e.g., MP3, Opus, Vorbis) because they correspond to common real-world compression pipelines and allowed us to study robustness under realistic deployment conditions (Table 2). Following the reviewer’s suggestion, we additionally evaluated the pretrained SONAR model on the In-the-Wild OOD test set after processing it with the DAC neural codec (16 kHz, encoding and then decoding), without retraining. We observe only moderate degradation: EER increases from 6.00% to 8.6%, while AUC remains high (0.977 VS 0.953). Importantly, score ranking is largely preserved (Spearman with rho = 0.944), with a low label-flip rate of 6.2%. These results suggest that SONAR captures structured LF-HF relationships rather than relying solely on fragile high-frequency artifacts. We agree that broader evaluation on additional neural codecs such as EnCodec and Mimi would further strengthen the paper, and we will pursue this in future work.
>
> Q2: How well does the method generalize to deepfake models that were not seen during training?
>
> We agree that this is a central question. SONAR is trained on ASVspoof2019 and evaluated on both ASVspoof2021 DF and the In-the-Wild dataset, both of which contain spoofing methods not seen during training. In particular, In-the-Wild was explicitly proposed to assess cross-model and cross-domain generalization. As shown in Table 1, SONAR achieves state-of-the-art performance on both benchmarks. In addition, our cosine-similarity analysis (Figure 4) and spectral probing experiment (Figure 5), both conducted on In-the-Wild, show that SONAR preserves discriminative LF-HF structure under distribution shift. These results support the claim that the proposed alignment mechanism improves robustness to unseen generation methods.

---

> > ### Author Rebuttal · Reviewer_ptqn · 2026-04-03
> >
> > I consider my main concerns to be fully resolved and I'll increase my score accordingly.

---

### Decision · Program_Chairs · 2026-04-30

**Decision:**

Accept (regular)

**Comment:**

This paper introduces SONAR, a method for audio deepfake detection that models the relationship between low- and high-frequency components, motivated by spectral bias in neural networks.

Reviewers agree the approach is well-motivated, clearly presented, and empirically strong, achieving state-of-the-art results on standard benchmarks while demonstrating robustness to codec and bandwidth variations. The idea of explicitly modeling LF / HF consistency is novel and supported by thorough ablations and analysis. Reviewers concerns such as robustness to modern neural codecs and generalization, were addressed during the rebuttal with additional experiments.

Overall, this is a solid and well-executed contribution with both practical relevance and novelty. I recommend acceptance.